# Social Support, Self-Care Behaviour and Self-Efficacy in Patients with Type 2 Diabetes during the COVID-19 Pandemic: A Cross-Sectional Study

**DOI:** 10.3390/healthcare9111607

**Published:** 2021-11-22

**Authors:** Premalatha Paulsamy, Rizwan Ashraf, Shadia Hamoud Alshahrani, Kalaiselvi Periannan, Absar Ahmed Qureshi, Krishnaraju Venkatesan, Vani Manoharan, Natarajan Govindasamy, Kousalya Prabahar, Tamilselvi Arumugam, Kumar Venkatesan, Kumarappan Chidambaram, Geetha Kandasamy, Rajalakshimi Vasudevan, Kalpana Krishnaraju

**Affiliations:** 1College of Nursing, Mahalah Branch for Girls King Khalid University, Abha 61421, Saudi Arabia; pponnuthai@kku.edu.sa (P.P.); shalshrani@kku.edu.sa (S.H.A.); 2Department of Pharmacology, College of Medicine and Dentistry, The University of Lahore, Lahore 55150, Pakistan; drriz72@yahoo.com; 3Oxford School of Nursing & Midwifery, Faculty of Health and Life Sciences, Oxford Brookes University, Oxford OX3 0FL, UK; kperiannan@brookes.ac.uk; 4Department of Pharmacology, College of Pharmacy, King Khalid University, Abha 62529, Saudi Arabia; aqureshi@kku.edu.sa (A.A.Q.); kumarappan@kku.edu.sa (K.C.); raja@kku.edu.sa (R.V.); 5Georgia CTSA, Emory University Hospital, Atlanta, GA 30322, USA; vani.manoharan@emoryheslthcare.org; 6Enhanced Mental Health Nurse, Elmmount Unit, St Vincent’s University Hospital, Elmpark, D04 T6F4 Dublin, Ireland; natrajan1975@yahoo.co.in; 7Department of Pharmacy Practice, Faculty of Pharmacy, University of Tabuk, Tabuk 71491, Saudi Arabia; kgopal@ut.edu.sa; 8Amity College of Nursing, Amity University Haryana, Gurugram 122412, India; atamilselvi@ggn.amity.edu; 9Department of Pharmaceutical Chemistry, College of Pharmacy, King Khalid University, Abha 62529, Saudi Arabia; kumarve@kku.edu.sa; 10Department of Clinical Pharmacy, Faculty of Pharmacy, King Khalid University, Abha 62529, Saudi Arabia; glakshmi@kku.edu.sa; 11Department of Pharmacy, Erode College of Pharmacy, Veppampalayam, Erode 638112, India; rnkkalpana@gmail.com

**Keywords:** Multidimensional Scale of Perceived Social Support (MSPSS), self-efficacy, self-care activities, social support, Summary of Diabetes Self-Care Activities (SDSCA) scale, HbA1c control, Diabetes Management Self-Efficacy Scale (DMSES)

## Abstract

Diabetes mellitus is a major public health issue that considerably impacts mortality, morbidity, and healthcare costs worldwide. The COVID-19 pandemic has created havoc in diabetes management, too, like other spectrums of life. A descriptive, cross-sectional study was adopted to determine the effect of Social Support, Self-Care Behaviour and Self-Efficacy in Type 2 Diabetes Mellitus (T2D) during this COVID-19 pandemic. Two hundred T2D patients who satisfied the inclusion criteria were chosen using a convenient sampling procedure. The tool consists of four sections, including socio-demographic characteristics, Multidimensional Scale of Perceived Social Support (MSPSS), revised Summary of Diabetes Self-Care Activities (SDSCA) Scale and modified Diabetes Management Self-Efficacy Scale (DMS). Descriptive and inferential statistics were used to analyze the obtained data. The mean and SD of diabetic management self-efficacy is 5.74 (1.95) and 4.37 (1.4), respectively, for patients with HbA1c < 6.5% and HbA1c ≥ 6.5%. The self-care activities of the patients who had good glycemic control were 4.31 (2.06) compared to 3.50 (1.73) who did not. The social support received by the patients was 6.13 (2.13) vs. 5.31 (1.67) among patients with glycemic control vs. no control. The results show that social support (*p* = 0.04), self-efficacy (*p* =0.01) and self-care activities (*p* = 0.001) were significantly related to the level of glycemic control of the T2D patients. A significant relationship was also identified between gender (*p* = 0.036), age (*p* = 0.001) and education status (*p* = 0.000) with HbA1c control of the participants. This study demonstrates a significant relationship between social support, self-care behaviours, self-efficacy and glycemic management in T2D patients. During this COVID-19 pandemic, interventions to enhance the self-care activities like exercise and social support to boost their self-efficacy; for better diabetes management, reducing diabetes complications or prolonging their onset are the need of the hour.

## 1. Introduction

Diabetes Mellitus (DM) is a chronic metabolic disorder characterized by high blood glucose levels. It is one of four non-communicable diseases (NCDs) on which world leaders are focusing. It is one of the most serious health problems of the twenty-first century, affecting about almost 500 million people worldwide, with that number expected to rise by another 30% by 2045 [1]. In India, around 50.9 million citizens are diabetic, and by 2025, India will be the diabetes capital of the world, with an estimated 80 million people diagnosed with the condition [2]. India already has the highest estimated number of individuals diagnosed with diabetes in the world (about 77 million in 2019), implying that India is home to one in every five diabetics worldwide [3]. Estimates from varied databases show a relatively high burden of diabetes and pre-diabetes in both rural and urban India, with a narrowing urban-rural gap. In the most recent decade (2010–2019), Goa (17.4 percent) and Tamil Nadu (24.0 percent) had the highest rural and urban prevalence rates, respectively. As a result, it is critical to develop immediate primary and secondary prevention strategies to limit further spread in high-prevalence areas [4]. Diabetes is putting a strain on India’s already strained healthcare system and economy [5].

In addition to placing a significant financial burden on individuals and society, type 2 diabetes (T2D) is the sixth leading cause of morbidity and death, affecting more than 4 million populace each year, with almost 10% of global all-cause mortality in the 20–99- year-old age range [1].Diabetes is also the fourteenth primary cause of Disability Adjusted Life Years (DALYs) in the globe [2] which accounted for at least US$ 760 billion in direct health expenditures in 2019 [1]. In India, one person dies every ten seconds from DM-related causes. Diabetes and cardiovascular disease have cost India $336 billion in the last national GDP. Between 2005 and 2015, $6 billion was spent [1]. T2D reduces the life expectancy of individuals diagnosed with the condition by around 15 years, making it a severe global problem [6].

There is a well known relationship between T2D and Cardio-Vascular (CVD) issues; more than half of diabetes patients die from CVD complications, particularly the younger population. Diabetes-related cardiac disorders account for more deaths in India than HIV/AIDS, Malaria, and Tuberculosis combined. Indians have the more significant affinity to developing a metabolic syndrome called “Syndrome X”, which makes them prone to elevated blood pressure, blood sugar and deranged blood lipids. According to a new study, Indians’ bodies are more insulin-resistant, makingthem at risk for T2D. For any given BMI, Joshi [4] found an Asian Indian phenotype, with a more significant percentage of body fat and a higher waist-to-hip ratio, which predisposes to diabetes and metabolic syndrome. As Indians have the highest tendencies to develop DM among all races, several challenges in diabetes management must be addressed, including rising prevalence, micro and macro-vascular complications, lifestyle modifications, delayed diagnosis, insufficient knowledge and soaring treatment costs. Hence, for combating this, Indians need to fight the hardest.

Preventing or delaying the onset of T2D can be accomplished with a balanced, nutritious diet, regular physical exercise, sustaining a healthy body weight, and keeping away from tobacco use. In addition to this, the right medications, regular screening and management for complications can all help to treat diabetes and delay or prevent the complications [5]. One of the significant aspects of DM care is perceived self-efficacy, which could lead to self-management behaviours and improve confidence among diabetic patients that they can manage diabetes. Moreover, the majority of diabetes’ care is provided by the self. Despite the excellent impact of self-care habits on chronic diseases, various studies have investigated the significance of disease control by patients themselves and indicated that self-care behaviours influence glycemic control [7,8]. Self-care refers to a patient’s utilization of their knowledge and skills to engage in healthy habits. Healthy dietary habits, physical activity, blood glucose self-monitoring, medicine and foot care are a few self-care practices [9].

Furthermore, social support, a societal component, has been proven to influence self-care and glycemic management in previous studies [10,11,12]. As a result, according to the American Dietetic Association (ADA), the majority of the patients require continuing diabetic self-management support (DSMS) to sustain self-management behaviours at the levels necessary to efficiently control diabetes [13].

T2D, and its related sequelae have recently become more common as a result of a sedentary lifestyle, especially during this pandemic where most of the world population is restricted in physical activity with social distancing, lockdown, etc. In 2020, a mysterious virus known as SARS-CoV-2 challenged the medical community and wrought havoc all over the globe. According to recently published data, COVID-19 patients have been prioritized over other clients, including DM patients, due to the unanticipated circumstances of the pandemic. Hence, the current study aimed to explore the effect of social support, self-care behaviours, and self-efficacy on glycemic control among T2D patients during this COVID-19 pandemic in India. Our findings provide additional practical guidance for healthcare practitioners on how to enhance diabetes control and promote and motivate healthcare practitioners to develop social support.

## 2. Materials and Methods

### 2.1. Design

A descriptive, cross-sectional study was adopted to determine the effect of Social Support, self-care behaviour, and Self-Efficacy in patients with type 2 diabetes during the COVID-19 pandemic.

### 2.2. Population and Setting

Patients with Type 2 Diabetes Mellitus were the study population. The data collection took place in the chosen diabetic clinics of a Corporate Hospital, Chennai, India.

### 2.3. Sample Size and Sampling Process

By the non-probability convenient sampling technique, 200 samples were selected from specific DM clinics of Corporate Hospital, Chennai, India. There were around 407 Type 2 Diabetes Mellitus patients enrolled in the diabetic clinics who regularly attended the follow-up. The minimum suggested sample size was 198, computed by Raosoft online sample size calculator having an overall population of 407 patients, a 95% confidence with a margin of error of 5%. Patients with active clinical record aged >18 years, able to communicate verbally, not hospitalized during the data collection, minimum of 1 year since the diagnosis, and with no mental retardation or other psychological issues such as mood and anxiety disorders were included in the study.

### 2.4. Data Collection Tools/Instruments

The self-administered tool consisted of 4 sections;

Section A: The socio-demographic characteristics of the Participants.

Section B: To measure social support, the Multidimensional Scale of Perceived Social Support (MSPSS) by Zimet et al. [14], was used, and earlier investigations have proven its reliability and validity [15,16]. The MSPSS questionnaire has 12 questions on a Likert scale of 0 to 7, with upper scores signifying more substantial social support.

Section C: The revised Summary of Diabetes Self-Care Activities (SDSCA) scale [17] measures the rate of self-care activity in the previous seven days (“days per week”) on six aspects of the diabetes routine: general eating pattern (healthy diet), specific diet, foot care, blood glucose monitoring, medication, physical activity and smoking status. It comprises 12 questions with scores ranging from 0 to 7 on a Likert scale; the high the score, the better the self-care.

Section D: The modified Diabetes Management Self-Efficacy Scale (DMS), with 20 items, was used to measure self-efficacy. It determines how confident participants are in their ability to deal with nutrition, blood sugar monitoring, foot inspections, physical activity, body weight, and medical management. Patients assessed themselves on an 11-point scale, with “zero” representing “cannot do at all” and 10 representing “certainly can do.” The participants’ self-efficacy was assessed by the mean scores of the 20 statements.

The tool was initially in English and translated into Tamil. The language validity of the questionnaire was determined by retranslating the Tamil version into English and ensuring that the actual meaning of each question was maintained (back-translation). The back-translation was done by the authors, who are competent in English and Tamil. The questionnaire was written in Tamil and disseminated in that language.

The instruments were tested with 20 patients in a different setting as a pilot project to see how well they were translated and how easy to administer. To verify understanding, participants were asked to read each item. Though it is a standardized tool, the content validity of the tool was obtained from five experts in Nursing and one general physician and a DM specialist. The reliability of the tool was assessed after the pilot study by Cronbach’s Alpha test (internal consistency), which was highly reliable. Each patient completed the tool in the presence of one of the investigators or a nurse skilled in data collection, and clinical data such as co-morbidities, diabetes duration, and HbA1c were collected from each patient’s current electronic health records.

### 2.5. Ethical Consideration

Official permission from the Medical Director of the hospital and ethical permission was obtained from the Institutional Ethical Committee with IEC/LCN/2021-11. Consent from the participants were collected before starting the study by explaining the aim of the study, their Role, confidentiality of the information and their right to depart from the study at any point of data collection.

### 2.6. Statistical Analysis

The data were processed and analyzed by SPSS 17.0 software using descriptive and inferential statistics. Descriptive statistics, χ^2^ tests (for categorical variables) and regression analysis to compare T2D patients’ self-efficacy (DMSES), self-care activities (SDSCA), social support (MSPSS) with HbA1c as well as demographics were performed and *p* < 0.05 is considered significant.

## 3. Results

The collected data from 200 patients were tabulated, and it was found that the information of 16 patients was incomplete. Therefore, data from only 184 participantswere analyzed and rounded as a percentage. The socio-demographic features of the study participants were tabulated in Table 1. According to this, most of the samples were females (65.76%), 82.06% of the participants married, and the remaining 17.94% were unmarried or without a spouse. More participants (45.65%) had DM for 6–10 years, and 79.89% of them had co-morbidities related to DM. Regarding the educational status, 19.02% had no formal education, 41.3% had studied up to high school, and 16.85% had graduate and above degree. The mean BMI was 30.07 with SD of 5.13, and the mean Glycosylated haemoglobin (HbA1c) was 6.91 ± 1.14, ranging from 5.1 to 11.3%. The occurrence of normal glycemic control (HbA1c < 6.5%) was 63.04% among the study participants.

Table 2 shows the DM Self-Care Activities of the T2D patients, which reveals that the patients follow the general diet in their day-to-day to life, followed by a specific diet for DM. On foot care and glucose monitoring, they had amean knowledge of 4.02 (1.17) and 4.27 (2.11), respectively. The score on exercise was the lowest compared to other aspects. Among the participants, 25.54% were smoking and the overall score of the participants on DM diabetes self-care activities was 4.56 with SD of 2.61 (Figure 1).

Table 3 demonstrates the regression analysis of HbA1c control with social support, self-care activities, diabetes self-efficacy and selected patient characteristics of the present study. According to the table, the social support (*p* = 0.000), diabetic self-care activities (*p* = 0.005) and diabetes management self-efficacy (*p* = 0.02) of the participants were positively associated with the glycemic control of the participants. A significant relationship was also observed between gender (*p* = 0.036), age (*p* = 0.001), as well as with education status (*p* = 0.000) and HbA1c level of the participants. Other demographic variables of the T2D patients did not show any significant relationship with their level of HbA1c control.

Table 4 shows the relationship between variables such as social support, self-care behaviour and diabetes self-efficacy, stratified with glycemic control. According to these findings, the diabetes self-efficacy and self-care activities of the T2D patients were strongly associated with their glycemic control at *p* < 0.01 and *p* < 0.001 levels of significance, respectively. The social support too has a significant positive relationship with the glycemic control of the T2D patients at *p* < 0.05 level. This shows that these variables have a strong association with the diabetes management of the patients, hence to be given importance while giving care to them.

## 4. Discussion

This study was performed to determine the effect of social support, self-care behaviour, and self-efficacy on glycemic control in T2D patients during this COVID-19 pandemic. Patients with diabetes need self-management and self-efficacy in order to prepare for nutrition, blood sugar control, physical activity, medications, and overall diabetes management [17]. The lack of self-efficacy in an individual with this chronic disease will have a detrimental influence on their quality of life since self-efficacy helps them manage and control their health by themselves [18]. Self-efficacy is a person’s belief in his or her ability to plan and carry out certain actions to accomplish specific outcomes. Self-efficacy is the ability to make decisions and commit to following them. A patient with high self-efficacy can be expected to lead a good quality of life in the long or short term [19]. There is a well established link between self-efficacy and blood sugar levels (BGL) and HbA1c levels in diabetic individuals and also correlated to quality of life [20,21].

According to the present study, most of the participants were females, married and had DM for 6–10 years with co-morbidities. Around 63% of the study participants were in normal glycemic control (HbA1c < 6.5%). Similar findings were reported in a study on the association between perceived social support and self-care behaviour in T2D patients, which showed that the females (67.4%) were predominant, and the majority (93.5%) of the participants were married [22,23]. The reason for having more female patients may be that there is an assumption that naturally, females are more compliant with the treatment regime and follow up their health status. In the same way, married patients may have family support to adhere to regular medical check-ups and compliance with the treatment regime.

According to the study by Gao, J. et al. [24], diabetes lasted an average of 8.3 years (SD = 6.4), whereas 69.8% said they experienced co-morbidities like hypertension, cardiovascular disease, stroke, or kidney disease, among other co-morbidities, which is analogous to the present study in which 79.89% of the participants had co-morbidities. In addition, the majority of them (70.7%) had an above-normal BMI with central obesity and had HbA1c levels that were considerably more elevated than those of healthy people. The ability to successfully manage diabetes daily is critical to achieving positive health outcomes. A sense of self-efficacy, or confidence in one’s self-management abilities, is essential to managing chronic diseases like DM successfully. Micro- and macrovascular problems also decrease as diabetic control is accomplished [25]. Hence, it is essential to continue reinforcing patients’ confidence in self-management either through family support or Patient-Provider Communication.

A study on the effect of the diabetes education and self-management (DESMOND) programme among newly diagnosed type 2 diabetes: a cluster randomized controlled trial in UK [26] revealed that the mean BMI was ~32 and mean baseline HbA1c was 7.9% in control and 8.3% in the intervention group which is more or less similar to our study findings. As it is well known that obesity is one of the major causes of T2D, the patients should be given constant motivation by the health care providers, family members, and significant others to bring lifestyle modifications such as regular exercise and dietary management. Similarly, only 63.04% of participants had a normal range, in the present study, regarding glycemic control. A study conducted in the United States revealed that inadequate glycemic control is associated with a prolonged duration of diabetes. It hypothesized that patients’ failure to achieve the optimal amount of glycosylated haemoglobin over time causes frustration and disappointment, lowering their self-efficacy. The duration of diabetes affects glycemic control in patients, according to the findings of the Trief et al. study [27]. Though in this study, the duration of DM did not show any significant association with HbA1c level, 37% of patients had uncontrolled DM, which may be due to the maximum of the study participants (72.28%) having DM more than 5 years. As it is cyclic, that is, due to prolonged duration, the self-efficacy of DM decreases and decreased self-efficacy leads to uncontrolled DM. A strong emphasis on enhancing the patient’s self-efficacy is to be given to bring substantial improvement in self-care management and control of diabetes.

The SDSCA scale score reveals that they have the better grades in general diet followed by specific diet for DM. The score on exercise was the lowest compared to other aspects, and 25.54% were smoking with an overall score of 4.56 with SD of 2.06 (Table 2). This result was in concurrence with the previous research [22] indicating that the self-care score had a mean and standard deviation of 4.31 ± 2.7, in which the exercise item had the lowest mean score of 3.38 ± 2.69 compared to other variables. This may be because though the participants know the value of regular exercise, they lack motivation and support from significant others like family. Moreover, most of the study participants were female, and they may not have time to do the exercise due to household responsibilities. In addition, due to COVID-19 and its related restrictions, the participants might not have had the opportunity for outdoor exercise. Hence, it is mandatory to provide appropriate and continuous motivation to the T2D patients, including their families, regarding the expected self-care activities to promote health and prevent DM-related complications. Moreover, more innovative in-door exercise strategies for T2D patients, such as aerobic exercises and yoga, to be taught to improve mobility, flexibility and burn calories.

The regression analysis shows that diabetes’ management self-efficacy (*p* = 0.02), diabetic self-care activities (*p* = 0.005) and social support (*p* = 0.000) received by the participants were significantly associated with the level of glycemic control of the participants (Table 3). These findings support prior research findings that self-care activities, diabetes Self-Efficacy (DMSE) and social support improve glycemic control of T2D patients [22,23,24,25,26,28]. This study found that social support and diabetes management self-efficacy are the remarkable predictors of DM self-care behaviour and thus for controlling diabetes.

In the present study, a significant relationship was also observed between gender (*p* =0.036), age (*p* = 0.001) and education status (*p* = 0.000) of the participants with their level of HbA1c control. This may be because men perceived more social support than women. After all, women have the greater obligation to care for the entire family, yet they may not receive the same social support as males. This social support helped men to follow their self-care activities and might have improved their DMSE. Similar findings were reported in a few studies [29,30]. As educational status plays an essential role in adherence to any treatment regime, in this study also, the educational status of the participants has a significant influence on their glycemic control. Regarding the age of the participants, generally, as the age advances, people acquire more maturity and their compliance augments. Moreover, while getting older, an individual’s perception towards health and the importance of maintaining it changes positively; therefore, they try to adhere to the health care regime, which is evident in the present study results.

According to the present study findings, diabetes management, self-efficacy and self-care activities were strongly associated with glycemic control at *p* ≤ 0.01 and *p* < 0.001, respectively. The social support also has a positive relationship with the glycemic control of T2D patients at *p* < 0.05 level. Our findings show that patients who have a firm belief in their abilities to control their condition are competent at doing so. The more critical research inquiry was whether diabetes management self-efficacy, patients’ self-care and social support have a substantial impact on diabetes control or not, which is accurate and was also consistent with other previous study findings [22,23,25]. This study has some limitations that should be acknowledged. First, our findings are the most relevant to the demographics studied; yet, most of the participants were selected from the diabetic clinics of a selected Corporate Hospital. Thus, future studies can be done in primary care settings or tertiary care hospitals to generalize the findings. This was a cross-sectional study and did not find any cause and effect relationship with the variables studied.

## 5. Implications

These study results indicate a significant association between social support self-care activities and diabetes self-care efficacy with diabetes management. This suggests that health care practitioners should pay attention to these aspects when treating a T2D patient and the family, as well as there is a need for innovative, informed diabetes management education during the ongoing COVID-19 pandemic.

## 6. Conclusions

To our knowledge, this is one of the first study to look at the impact of social support, diabetes self-care behaviour/activities and diabetes management self-efficacy on blood glucose control in the Indian population with T2D during the COVID-19 pandemic. We found that these variables have a significant positive association with each other. Hence, appropriate, continuing education and motivation are needed to improve self-efficacy, self-care activities to the diabetic patients and including their families in these aspects, especially innovative strategies to improve their physical activity during the pandemic.

## Figures and Tables

**Figure 1 healthcare-09-01607-f001:**
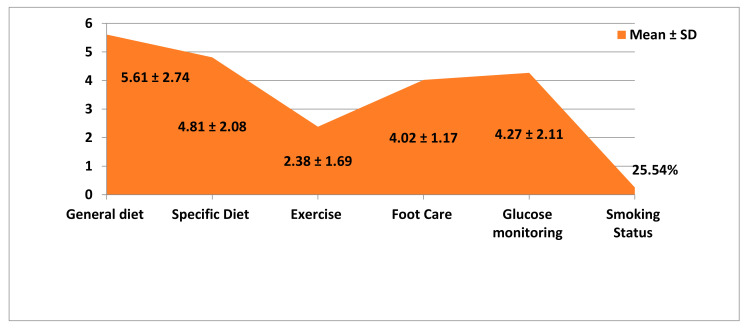
Mean ± SD of Self-Care Behavior of the T2D patients.

**Table 1 healthcare-09-01607-t001:** Specifications of stories of building.

Socio-Demographic Characteristics	No.	%
Age (Years)		
26–35	8	4.35
36–45	61	33.15
46–55	73	39.67
56–65	29	15.76
>65	13	7.07
Gender		
Male	63	34.24
Female	121	65.76
Marital status		
Married	151	82.06
Unmarried	33	17.94
Time since diagnosis of diabetes		
1–5 years	51	27.72
6–10 years	84	45.65
>10 years	49	26.63
Comorbidity		
Yes	147	79.89
No	37	20.11
Education		
No formal education or elementary school	35	19.02
High school	76	41.30
Higher secondary school	42	22.83
Graduate and above	31	16.85
Employment status		
Unemployed or no organized employment	67	36.41
Government employee/Army/Police	44	23.91
Private employee	17	9.24
Self Employed	38	20.65
Retired	18	9.78
Glycemic Control		
HbA1c < 6.5%	116	63.04
HbA1c ≥ 6.5%	68	36.96
Mean BMI (kg/m^2^) ± SD	30.07 ± 5.13	

**Table 2 healthcare-09-01607-t002:** The DM self-care activities of the T2D patients.

Characteristics	Mean ± SD
General Diet	5.61 ± 2.74
Specific Diet	4.81 ±2.08
Exercise	2.38 ± 1.69
Foot Care	4.02 ± 1.17
Glucose Monitoring	4.27 ± 2.11
Smoking Status	25.54%
Summary of Diabetes Self-care	4.56 ± 2.61

**Table 3 healthcare-09-01607-t003:** Regression analysis of HbA1c control with social support, self-care activities, diabetes self-efficacy and selected patient characteristics.

Variables	Beta Coefficient	*t*-Value	*p*-Value	95% CI	
				Lower	Upper
Social Support	0.34	4.5	0.000 ***	0.24	0.61
DM Self-care Activities	0.36	2.91	0.005 **	1.54	0.29
DM Self Efficacy	0.26	2.38	0.020 *	1.34	0.12
Gender	0.23	2.13	0.036 *	1.86	0.06
Age	0.26	3.6	0.001 ***	0.09	0.32
Education	0.49	5.88	0.000 ***	0.26	0.52

* *p* < 0.5, ** *p* < 0.01, *** *p* < 0.001.

**Table 4 healthcare-09-01607-t004:** Relationship between variables stratified by glycemic control status.

Variables	Glycosylated Hemoglobin < 6.5%	Glycosylated Hemoglobin ≥ 6.5%	*p*-Value
Number of patients	116	68	
MSPSS (Mean (SD))	6.13(2.13)	5.31(1.67)	0.042 *
SDSCA (Mean (SD))	4.31(2.06)	3.50(1.73)	<0.001
DMSES (Mean (SD))	5.74(1.95)	4.37(1.4)	<0.01

MSPSS-Multidimensional Scale of Perceived Social Support; SDSCA- Summary of Diabetes Self-Care Activities (SDSCA) scale; DMSES- Diabetes Management Self-Efficacy Scale. * *p* < 0.5.

## Data Availability

The data presented in this study is available on request from the corresponding author.

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
