# Peer review of "Social Support, Self-Care Behaviour and Self-Efficacy in Patients with Type 2 Diabetes during the COVID-19 Pandemic: A Cross-Sectional Study"

_healthcare, 2021, doi:10.3390/healthcare9111607_

Round 1

Reviewer 1 Report

The title of the article suggests that the paper focuses on the impact of the pandemic on social support, self-care behavior, and self-efficacy in patients with type 2 diabetes during the COVID-19 pandemic, however, throughout the paper the study appears not to focus on the impact of the pandemic on the health status of diabetic patients.

For example, the introduction could deal more with the overall impact that COVID-19 has had on health care. If the condition assessed in this article is related to the pandemic, the introduction could be nurtured by talking more about the health care landscape during the health crisis and the impact of COVID-19 on public health.

The objective of the paper is interesting for public health and it is a topic to be considered due to the health crisis resulting from COVID-19, so this paper can have more impact if the groups to be evaluated are rethought, one suggestion is to divide the patient population into those who have difficulties in developing self-care behaviors and who have social support problems and another population who do not have these deficiencies.

This type of work has great potential because it can highlight the social and public health crisis derived from the COVID-19 pandemic, however, it is suggested that the authors rethink the design of the work to get the most out of it and thus have a great impact.

Author Response

The title of the article suggests that the paper focuses on the impact of the pandemic on social support, self-care behavior, and self-efficacy in patients with type 2 diabetes during the COVID-19 pandemic, however, throughout the paper the study appears not to focus on the impact of the pandemic on the health status of diabetic patients.

For example, the introduction could deal more with the overall impact that COVID-19 has had on health care. If the condition assessed in this article is related to the pandemic, the introduction could be nurtured by talking more about the health care landscape during the health crisis and the impact of COVID-19 on public health.

Included in the introduction

The objective of the paper is interesting for public health and it is a topic to be considered due to the health crisis resulting from COVID-19, so this paper can have more impact if the groups to be evaluated are rethought, one suggestion is to divide the patient population into those who have difficulties in developing self-care behaviors and who have social support problems and another population who do not have these deficiencies.

The suggestion given by the reviewer is well taken and unfortunately, this study did not differentiate the groups with or without difficulties, it was not included in the manuscript. Indeed, this will be taken due consideration in the future projects.

This type of work has great potential because it can highlight the social and public health crisis derived from the COVID-19 pandemic, however, it is suggested that the authors rethink the design of the work to get the most out of it and thus have a great impact.

The suggestion given by the reviewer is well appreciated and unfortunately, this study’s scope was limited to the effect of Social Support, Self Care Behaviour and Self-Efficacy in patients with type 2 diabetes during the COVID 19 pandemic and a cross sectional study, it was not included in the manuscript. Indeed, this will be taken due consideration in the future follow up projects.

Reviewer 2 Report

This is the first studies to look at the impact of social support, diabetes self-care behaviour/activities and diabetes management self-efficacy on blood glucose control in the Indian population with T2D during the Covid -19 pandemic.

The study suggests that the health care practitioners should pay attention on these aspects when treating a T2D patient and his family and there is need for informed diabetes management education to them during the ongoing COVID-19 pandemic period.

This article is well written and of clinical interest.

However, several issues should be improved before the consideration for publication.

1 MSPSS, SDSCA, and DMSES are not so much familiar to the general readers. Please provide the  brief explanation for interpretation to the sentences in Line 137-152.

2 It may be better to show also the pre-pandemic data.

3 I would like to know the changes of the oral anti-diabetic drugs during study. 

4 I am interested in cognitive behavioral therapy. I would like you to mention cognitive behavioral therapy in your discussion.

Author Response

Comments and Suggestions for Authors

This is the first studies to look at the impact of social support, diabetes self-care behaviour/activities and diabetes management self-efficacy on blood glucose control in the Indian population with T2D during the Covid -19 pandemic.

The study suggests that the health care practitioners should pay attention on these aspects when treating a T2D patient and his family and there is need for informed diabetes management education to them during the ongoing COVID-19 pandemic period.

This article is well written and of clinical interest.

However, several issues should be improved before the consideration for publication.

  1. MSPSS, SDSCA, and DMSES are not so much familiar to the general readers. Please provide the  brief explanation for interpretation to the sentences in Line 137-152.

                        Explanations given

  1. It may be better to show also the pre-pandemic data.

                        Included in introduction (ref 4)

  1. I would like to know the changes of the oral anti-diabetic drugs during study. 

Medication adherence was one of the component included in the Summary of Diabetes Self-Care Activities (SDSCA) scale and this was not measured in isolation as it was not the scope of the study. The study measured only the effect of social support, self care activities and diabetes self efficacy on Glycemic control of the patients. The concern of the reviewer is well appreciated and it will be given due consideration in the future project.

  1. I am interested in cognitive behavioral therapy. I would like you to mention cognitive behavioral therapy in your discussion.

Apologies for not able to include the cognitive behaviour therapy as the investigators are not qualified on the same. This requires special training and preparatory certification to incorporate/provide CBT for patients. It will be, indeed, considered in future endeavours.

Reviewer 3 Report

In this paper (Manuscript ID healthcare-1422524) Premalatha Paulsamy and co-authors have carried out a descriptive, cross-sectional study to determine the effect of Social Support, Self-Care Behaviour and Self-Efficacy in Type 2 Diabetes Mellitus during this COVID -19 pandemic. The idea to relationship among these 3 parameters and management of diabetes during pandemic is interesting.

In general, the manuscript is clear, easy to read and understand. However, although the experimental design is appropriate any details should be improved in material and methods, result and discussion sections.

Comments for consideration:

-Line 172: In this section the authors should indicate the significant value for p-value.

-Line 197: Table 3 should have a table foot which should indicate the significant value for *; **; and *** p-value.  

-Line 208: Figure 1 is not referenced into the text. The authors should put a table foot where they should explicate what we are view and how it is has made.

-Line 209: Table 4 should have a table foot where the authors should show the significant value for p-value.

-Line 230-236: This paragraph is part of result, even it is written in Line 177-178. The authors should not include result in discussion.

-Line 277-281: This paragraph is part of result, even it is written in Line 191-196.  The authors should not include result in discussion.

-Line 294-297: This paragraph is part of result, even it is written in Line 199-202.  The authors should not include result in discussion.

-Line 302-303: This paragraph is part of result, even it is written in Line 203-206. The authors should not include result in discussion.

Author Response

Comments and Suggestions for Authors

In this paper (Manuscript ID healthcare-1422524) Premalatha Paulsamy and co-authors have carried out a descriptive, cross-sectional study to determine the effect of Social Support, Self-Care Behaviour and Self-Efficacy in Type 2 Diabetes Mellitus during this COVID -19 pandemic. The idea to relationship among these 3 parameters and management of diabetes during pandemic is interesting.

In general, the manuscript is clear, easy to read and understand. However, although the experimental design is appropriate any details should be improved in material and methods, result and discussion sections.

Comments for consideration:

-Line 172: In this section the authors should indicate the significant value for p-value.

Included

-Line 197: Table 3 should have a table foot which should indicate the significant value for *; **; and *** p-value.  

Included

-Line 208: Figure 1 is not referenced into the text. The authors should put a table foot where they should explicate what we are view and how it is has made.

Included

-Line 209: Table 4 should have a table foot where the authors should show the significant value for p-value.

Included

-Line 230-236: This paragraph is part of result, even it is written in Line 177-178. The authors should not include result in discussion.

Modified

-Line 277-281: This paragraph is part of result, even it is written in Line 191-196.  The authors should not include result in discussion.

Modified

-Line 294-297: This paragraph is part of result, even it is written in Line 199-202.  The authors should not include result in discussion.

Modified

-Line 302-303: This paragraph is part of result, even it is written in Line 203-206. The authors should not include result in discussion.

Modified

Round 2

Reviewer 1 Report

After reading the new version of the article with the applied changes and the explanation of the authors, it is suggested to publish the article. Primary care is the pillar of public health. This type of article is relevant because it emphasizes the importance of prevention, especially in patients with diabetes mellitus. It only remains to mention that some terms should be homogenized, such as "Covid" which appears in the article as "Covid" or "COVID", but, this item may be attended by the editors.